# Sky and wire: A UAV-BPL synchronization algorithm for information exchange in herding operations

Bashir Olaniyi Sadiq [1,2], Motirh Al-Mutairi[3], Yusuf Abubakar Sha'aban [4*],
Houssem Rafik El-Hana Bouchekara[5], Khalid Almutairi[6]

1 Department of Computer Engineering, Ahmadu Bello University, Zaria, Nigeria, 2 Department of
Electrical, Telecommunication and Computer Engineering, Kampala International University, Kampala,
Uganda, 3 Department of Geography and Environmental Sustainability, Princess Nourah Bint
Abdulrahman University, Riyadh, Saudi Arabia, 4 Department of Electrical, Computer and Software
Engineering, Embry-Riddle Aeronautical University, Prescott, Arizona, United States of America,
5 Department of Electrical Engineering, University of Hafr Al Batin, Hafr Al Batin, Saudi Arabia,
6 Mechanical Engineering Technology, University of Hafr Al Batin, Hafr Al Batin, Saudi Arabia

* shaabay1@erau.edu

org/10.1371/journal.pone.0334888

University, TANZANIA, UNITED REPUBLIC OF

**Peer Review History:** PLOS recognizes the
benefits of transparency in the peer review
process; therefore, we enable the publication
of all of the content of peer review and
author responses alongside final, published
articles. The editorial history of this article is
available here: https://doi.org/10.1371/journal.
pone.0334888

## Abstract

Herding operations in rural areas often lack reliable wireless connectivity, hindering
real-time monitoring of livestock, environmental conditions, and operational status.
Existing solutions like satellite or cellular networks are costly or impractical, while
short-range wireless technologies struggle with large farm coverage. This paper
presents a UAV-BPL synchronization algorithm that uses the farm's existing electrical
grid via Broadband over Power Line (BPL) communication to enable cost-effective,
scalable data collection. The algorithm integrates Unmanned Aerial Vehicles (UAVs)
with BPL relay points to optimize information exchange in herding operations. Two
operational scenarios are proposed in the paper: the Coverage-Driven scenario that
maximizes data collection from all sources, achieving 99.2% coverage and critical
data redundancy, and the Priority-Driven scenario that focuses on high-urgency data,
covering 100% of priority sources with ~12% energy consumption. The paper eval-
uates performance indicators such as coverage, energy usage, frequency of relay
visits, and response timeliness using MATLAB simulations. Results demonstrate
that the Coverage-Driven scenario excels in comprehensive monitoring, while the
Priority-Driven scenario ensures energy-efficient, rapid responses to critical events.
By combining BPL's robust connectivity with UAV mobility, the proposed method
enhances farm management, scalability, and adaptability, offering a practical solution
for rural dairy herding operations to improve productivity and sustainability.

## Introduction

Herding operations in rural areas face significant challenges in achieving real-time
monitoring of livestock, environmental conditions, and operational status due to

**Data availability statement:** The data underlying the results presented in the study are available from https://github.com/bosadiq/UAV-BPL-synchronization-algorithm.

**Funding:** Princess Nourah bint Abdulrahman University Researchers Supporting Project number (PNURSP2025R241), Princess Nourah bint Abdulrahman University, Riyadh, Saudi Arabia. The funders had no role in study design, data collection and analysis, decision to publish, or preparation of the manuscript.

**Competing interests:** The authors have declared that no competing interests exist.

limited wireless connectivity. Wi-Fi and cellular networks, which are conventional communication methods, are often unreliable or unavailable in remote locations because of sparse infrastructure and challenging terrain [1–3]. This leads to delays in detecting critical events, such as cattle illnesses or environmental changes. Thus, reducing farm productivity and sustainability [4,5]. While satellite communication offers an alternative, its high cost makes it inaccessible for small-to-medium scale farmers. On the other hand, short-range wireless technologies like ZigBee and Bluetooth are also inadequate for covering large farm areas, suffering from signal attenuation and high energy demands [6–10]. As such, many rural farms rely on manual monitoring, which can be labor-intensive and error-prone [11].

BPL communication emerges as a promising solution by leveraging existing electrical infrastructure to transmit data reliably over long distances [12,13]. BPL uses the farm's existing electrical power cables as a communication backbone for powering the barns, milking stations, and other farm buildings as a pre-made communication network that offers high bandwidth of up to 100 Mbps. This is in addition to providing weather-resistant connectivity without the need for costly new installations [14,15]. Nonetheless, BPL alone cannot address the dynamic data collection needs of large farms, as sensors are often distributed across remote pastures, inaccessible to fixed relay points. Therefore, it requires the use of UAVs. The availability of low-cost UAVs has made them accessible to individuals, including farmers. Hence, UAVs could be used in conjunction with the BPL to provide full coverage of the farms and herding fields. UAVs complement BPL by providing aerial mobility to collect data from dispersed sensors and deliver it to BPL relays, enhancing coverage and responsiveness [16,17]. Unlike direct BPL transmission to farm operators or the cloud, which lacks flexibility for real-time interventions, UAVs enable targeted data retrieval and imagery capture, critical for urgent scenarios like health alerts or weather events [18,19]. Fig 1 presents a typical layout of UAVs in an agricultural setting.

When combined, UAV and BPL create a dynamic framework for information sharing that redefines the way herding farms communicate, keep a surveillance on things, and react instantly [18–20]. In this situation, BPL communication stands out

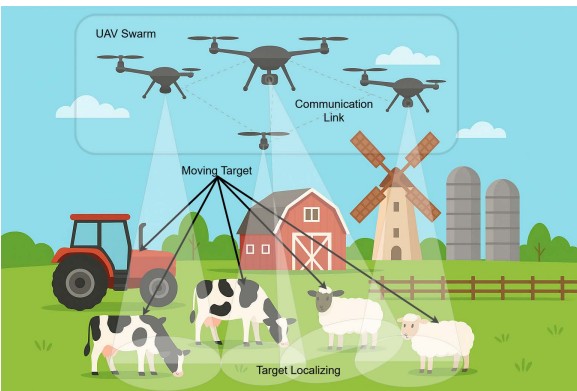

**Fig 1. UAV in agricultural environment [17].**

as a very effective solution, and its application in herding operations is clearly warranted. Despite advancements in smart agriculture, existing solutions face significant limitations. Satellite-based systems, while robust, are cost-prohibitive and complex to deploy [20]. Wireless protocols like LoRa and ZigBee offer low-power communication but are constrained by limited range and bandwidth. Making it unsuitable for high-volume data streams [21,22]. Recent studies, such as those exploring sustainable farming using AI and IoT technologies [23] and UAV networks for air patrol with a view to gathering information, and precisely respond to farm demand [24–26], highlight energy and scalability challenges, particularly for smallholder farmers. The concept of BPL technology is demonstrated in Fig 2.

The objective of the work is to develop a UAV-BPL framework for smart herd farming, integrating UAVs with high-bandwidth BPL communication to optimize real-time data collection, achieving low latency and high coverage while reducing manual labor and enhancing animal welfare through efficient, scalable monitoring compared to traditional wireless methods. This study proposes a UAV-BPL synchronization algorithm designed to optimize information exchange in dairy farm herding operations. By integrating UAVs with BPL relay points, the algorithm minimizes UAV travel distance while prioritizing the routing of urgent data, such as livestock vitals or environmental alerts. The proposed framework leverages BPL's stability to aggregate data from diverse sources such as cattle sensors and weather stations at fixed relay points, which UAVs access efficiently, reducing energy consumption and enhancing scalability. The main contributions of the paper are highlighted as follows:

1. The paper presents a novel UAV-BPL sites synchronization method, which minimizes UAV travel distances and maximizes information sharing.

2. The proposed method offers a robust, cost-effective substitute for conventional connectivity methods in remote locations by combining BPL's utilization of existing power lines with UAV agility.

3. A flexible, priority-driven approach that addresses diverse farm data types, surpassing prior efforts focused solely on livestock health.

The paper is structured as follows: the first section introduces the paper. The second section discusses related works and explores the fundamental concepts of UAV and IoT in smart agriculture. The third section presents the methodology and algorithm, while the fourth section presents the results and discussions. Finally, the fifth section presents the conclusion to the work.

## Related work

Advancements in Internet of Things (IoT) devices, UAVs, sensors, and communication technologies such as Flying Ad Hoc Networks (FANETs) and low-power wireless systems have resulted in their adoption in smart farming. These

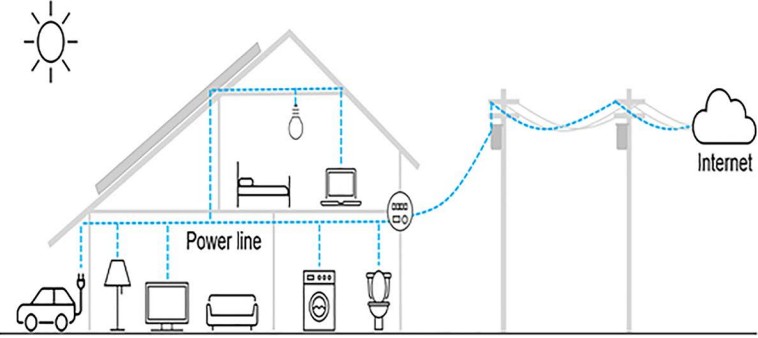

**Fig 2. Typical concept of the BPL technology [27].**

technologies support precision agriculture through precise data acquisition, interconnectivity, and efficient management practices in livestock farming, especially in expansive rural environments. However, limited wireless coverage and high energy consumption associated with mobile UAV operations remain critical challenges. This section reviews existing solutions, focusing on wireless technologies, satellite-UAV systems, and BPL applications, highlighting their limitations and the need for a cost-effective, scalable alternative.

The study reported in [26] proposes multi-objective UAV swarm optimization for multi-target localization in agricultural settings using Pareto Optimality Theory, balancing energy usage, communication latency, and task distribution. On the other hand [27], explores the coordination of UAVs and satellites within Space-Air-Ground Integrated Networks (SAGIN), using Age of Information (AoI) as a freshness metric and examining interference in Full Frequency Reuse (FFR) vs. frequency division schemes. One notable field-tested implementation is the (Space-Air-Ground integrated Grazing IoT) SAG-GIoT system demonstrated by [28], which uses BeiDou satellite messaging and (Long Range) LoRa in rugged pastoral zones of the Qinghai-Tibet Plateau. Though cost-effective and scalable, the system is constrained by LoRa's low data bandwidth, limiting its application to sensor updates rather than real-time multimedia or bulk data. This limitation presents a problem that could be well suited for high data bandwidth technologies such as BPL. BPL's role in smart grids has been extensively studied, especially in rural electrification contexts. Galli et al. [29] present how BPL functions both as the control and data transmission backbone in power infrastructure, supporting broadband speeds up to 100 Mbps over several kilometers. This positions BPL as a viable rural alternative to conventional approaches such as Wi-Fi, ZigBee, or LoRa. Moreover, BPL's robustness and weather resistance in harsh grid environments position it as a viable option for connectivity in agricultural settings. Ferreira et al. [30] and Lampe et al. [31] further show how BPL handles signal attenuation, impedance, and multiple access control schemes to deliver consistent service across large geographic areas, with minimal additional infrastructure costs due to reliance on existing power lines. These properties make BPL a highly compatible medium for use with UAV-based data retrieval in large pastures, barns, and remote sensors as required by our application.

Another work [32], introduces the Air-to-ground, Energy-aware mission-oriented protocol, leveraging the Fuzzy Logic (AERO-FL) routing protocol in a FANET with five UAVs and IEEE 802.11n. This protocol achieves real-time image transmission but at high energy costs and requires line-of-sight (LoS). Likewise, [33] leverages a CTracker model for cattle tracking, but the protocol relies on 802.11 wireless, which offers limited range and bandwidth compared to BPL, in addition to the unavailability of the wireless technology in some remote locations. Other studies explore ZigBee [34], Bluetooth [35], and hybrid wireless schemes in Flying Ad-Hoc Networks (FANETs). While suitable for energy-efficient UAV-to-UAV (U2U) and UAV-to-Infrastructure (U2I) communication, these technologies fall short in rural scenarios demanding long-range, high-bandwidth, and low-latency performance. For instance, [36] showed high packet loss using IEEE 802.15.4 over rural landscapes due to foliage and terrain. Researchers like [37] have proposed integrated satellite–aerial–terrestrial integrated network (SATIN) architectures with multi-UAV Internet of Remote Things (IoRT) deployments to address coverage, interference, and battery drain challenges. Although these systems increase network resilience and coverage, they also demand more capital and technical overhead, which is an obstacle for smallholder farmers. Therefore, the BPL-UAV synergy offers a practical compromise by centralizing data relay at power poles and allowing UAVs to retrieve batches of high-priority sensor data. BPL's long range, stability, and bandwidth overcome the limitations of short-range wireless technologies. The UAVs, in turn, avoid frequent battery-depleting flights, maximizing efficiency and scalability. Therefore, the literature supports using BPL as a communication backbone for smart agriculture in rural locations, especially when combined with UAVs for dynamic data collection and action. As summarized in Table 1, the UAV-BPL configuration outperforms traditional wireless-only systems across multiple operational dimensions.

A summary of some of the pertinent works is given in Table 2.

Based on the pertinent works of literature, key gaps are identified and are as follows: (1) low-bandwidth wireless protocols such as ZigBee, amongst others, fail to support high-volume data over large farms. (2) Satellite-based systems are cost-prohibitive for smallholder farmers to deploy. (3) UAV-based solutions suffer from high energy consumption and

**Table 1. UAV communication technologies comparison.**

| Aspect | UAV-Wireless (ZigBee/Wi-Fi/LoRa) | UAV-BPL (Power Line Communication) |
|---|---|---|
| Communication Range | Short (50–500 m) | Long (up to several km) |
| Data Rate | Low to Moderate (0.25–10 Mbps) | High (up to 100 Mbps) |
| Energy Consumption | High (battery-draining for UAVs) | Low (centralized relay points) |
| Interference Susceptibility | High (RF-based interference) | Low (uses shielded lines) |
| Infrastructure Requirement | Minimal (but requires clear LoS) | Uses existing power grid |
| Deployment Cost | Low initial cost | Moderate to Low (infrastructure reused) |
| Weather Resistance | Low (susceptible to rain, fog) | High (insulated against weather) |
| Scalability | Limited (dense deployment issues) | High (easily extended with grid) |
| Latency | Moderate to High | Low (wired reliability) |
| Best Use Case | Low-power short-range sensing | High-volume, wide-area farm coverage |

**Table 2. Summary of relevant works.**

| Citation | Methodology | Contributions | Limitations |
|---|---|---|---|
| [26] | Employed MOEA with POT for UAV swarm optimization, balancing task allocation, communication, and EC in MMT localization. | Achieved lower MSE, reduced EC, minimal collisions, and a latency of 11.97 ms; improved SF scalability and accuracy. | Lower data rate (40–90 MHz, less suited for static high-bandwidth tasks. |
| [27] | Satellite-UAV collaboration with FFR and FDMA strategies. | Derives closed-form AoI expressions for fused satellite-UAV data | Requires dedicated RF bands, unlike BPL's power-line infrastructure |
| [28] | Developed the SAG-GIoT system with a three-tier architecture using BeiDou Satellite and LoRa communication. | Introduced a low-cost, scalable IoT system for yak herding, enabling efficient daily supervision, UAV grazing, and yak retrieval, with flexible deployment in remote areas | Limited data bandwidth due to LoRa and BeiDou short messages; lacks detailed metrics on communication reliability and scalability |
| [32] | Deployed AERO-FL in a FANET with 5 UAVs, 25 ground nodes in a 5x5 grid, using IEEE 802.11n, fuzzy logic in NS-3 for real-time data transmission | Enabled real-time 2 MB image transmission every 5 seconds, reduced mission time, and optimized routing with dynamic power adjustments. | Lower data rate (0.5 Mbps vs. BPL's up to 200 Mbps), limited range (400 m vs. BPL's km-scale), higher energy use, and less suited for static, high-bandwidth applications. |
| [33] | Used enhanced CTracker with ResNet-18, FPN, hybrid deblurring, FCOSx detection, and ChainingX for real-time MOT on UAVs | Achieved 77 fps tracking, MOTA of 31.2%, IDF1 of 43.6%, improved occlusion handling, and accurate cattle tracking in real-time | shorter range (UAV-ground link), higher energy use, and less suited for static high-bandwidth tasks. |
| [36] | Used a testbed with a UAV, IEEE 802.15.4, to analyze A2G packet loss in rural fields. | Characterized the IEEE 802.15.4 A2G link with a 0.304 packet loss rate, and provided sensor deployment insights for smart farming. | shorter range of 50m, less suited for static high-bandwidth tasks. |
| [35] | Proposed a hybrid 802.15.1/802.11 communication scheme for FANET, | Improved throughput and reduced delay in FANET, enabled low-cost, energy-efficient UAV deployment with a multi-layer architecture | BPL can be used to improve the Lower data rate and shorter range of the communication infrastructure (UAV-to-Ground) used |
| [37] | Proposed a CPO algorithm with BCD and DDK for multi-UAV-assisted IoRT in SATIN, simulated over 800 m x 800 m with 30 devices. | Achieved higher uplink data rate, ensured full device coverage, optimized UAV deployment, device correlation, and power | Satellite links in terrestrial-satellite networks are vulnerable to environmental interference, such as atmospheric attenuation, rain fade, or ionospheric scintillation, which can disrupt ground-satellite connections |
| [34] | Used ZigBee protocol for multi-UAV | Achieved 84% success in image transmission, average time 62.9–64.8s, suitable for remote monitoring | Lower transmission data rate |

limited range. (4) Existing BPL applications lack dynamic data collection mechanisms. Therefore, the proposed UAV-BPL synchronization algorithm addresses these gaps by combining BPL's high-bandwidth, long-range connectivity with UAVs' aerial mobility. Unlike wireless systems, it leverages existing power lines for cost-effective, weather-resistant data transfer. Compared to satellite solutions, it reduces costs by using farm infrastructure. By centralizing data at BPL relays, it minimizes UAV travel and energy use, while its priority-driven approach ensures the timely delivery of critical data, enhancing scalability and adaptability for rural herding farms.

## Methodology

This section presents the methodology for the UAV-BPL synchronization framework, designed to optimize information exchange in dairy farm herding operations. The framework uses the farm's electrical grid via BPL communication, integrated with UAVs, and relay points to collect and process data from sensors. The methodology is structured in three phases: data collection, UAV-BPL synchronization, and information processing and exchange, with two operational scenarios (Coverage-Driven and Priority-Driven) to address diverse farm operational needs. The UAV-BPL coordination framework is depicted in Fig 3. The head-end is the equipment at the Internet Service Provider (ISP) side where the data from the power line network is collected and injected into the internet backbone or the cloud while the LV line is the Low Voltage line.

### Assumptions and system overview

The framework assumes a herding farm with existing electrical infrastructure supporting BPL communication that adheres to the IEEE 1901 standards for high-bandwidth data transfer (up to 100 Mbps) over power lines. The grid is assumed reliable, with noise levels below 40 dB and interference mitigated by BPL modems. Relay points, equipped with BPL modems & isolation circuits and low-power Wi-Fi modules, aggregate data from sensors and communicate with UAVs. Safety is ensured through isolation circuits and step-down transformers that protect IoT and other devices from high-voltage lines.

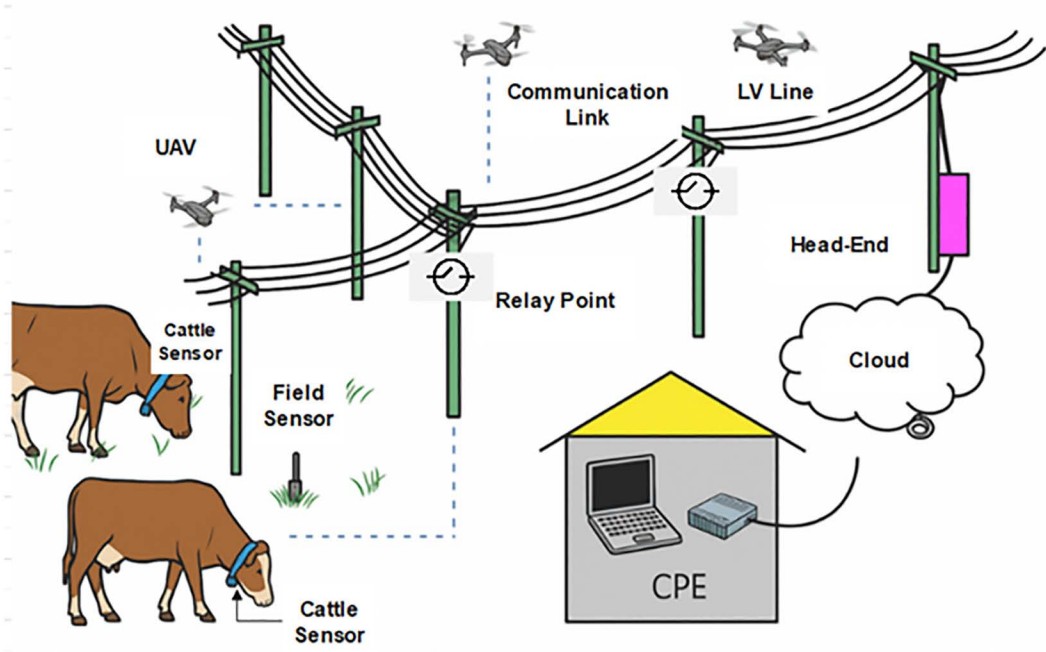

**Fig 3. UAV-BPL coordination framework.**

Thereby, minimizing risk to human operators and equipment [38]. Prioritizing computational simplicity, flight time for UAV data transport is neglected due to short distances between relays and UAVs. The system is scalable, supporting a larger number of UAVs and relay nodes used herein, with performance expected to degrade gracefully for larger farms due to increased coordination overhead. The table of symbols and notations is presented in Table 3.

**UAV-BPL synchronization**

Synchronization between UAVs and BPL relay points is achieved through a two-layer communication protocol (physical layer and medium access control layer). The BPL network maintains a channel capacity $C_{BPL}(t) \geq C_{min}$ to ensure reliable data relay. Relay points are presumed to use Wi-Fi (IEEE 802.11n) for short-range (<100m) communication with UAVs, employing a time-division multiple access schedule to avoid interference. Relay points are hardware modules with BPL modems, Wi-Fi transceivers, and microcontrollers, mounted on power poles.

The methodology for the proposed synchronization algorithm is broken down into three distinct phases as follows:

1. Data Collection: Sensors placed across the herding farm generate data packets $P_i(t)$ at time $t$, where $i \in \{1, \ldots N\}$ denotes the index of the sensor. Packets are transmitted via BPL to the nearest relay point along power lines. The relay points, positioned along power lines, serve as data hubs, consolidating information for UAV access. If the packet has a high priority $p_i(t) > \theta$, UAVs are notified to collect data directly from the sensor. Priority $p_i(t) \in [0, 1]$ is computed in real time based on the sensor algorithm evaluating critical thresholds.

2. UAV Synchronization: UAVs indexed $j \in \{1, \ldots M\}$ continuously monitor nearby sensors within their communication radius with the help of Wi-Fi (IEEE 802.11n) with a TDMA schedule to avoid interference. For each sensor $i$, the UAV evaluates energy levels and priority against predefined thresholds according to equations (1) and (2):

$$E_j(t) \geq E_{min} \tag{1}$$

$$p_i(t) \geq \theta) \tag{2}$$

**Table 3. Symbols and notations.**

| Symbol | Notations |
| --- | --- |
| N | Number of sensor nodes |
| M | Number of UAVs |
| $P_i(t)$ | Data packet from sensor $i$ at time $t$ |
| $d_{ij}(t)$ | Distance from UAV $j$ to relay point that aggregates data from sensor $i$ |
| $x_{ij}(t)$ | Binary decision variable (1 if UAV $j$ collects data from sensor $i$, 0 otherwise) |
| $E_j(t)$ | Energy level of UAV $j$ at time $t$ |
| $E_{min}$ | Minimum UAV energy threshold (20%) |
| $p_i(t)$ | Priority score of packets $i$ (0–1) |
| $\theta$ | Priority threshold |
| $C_{BPL}(t)$ | BPL channel capacity at time $t$ |
| $C_{min}$ | Minimum BPL capacity |
| $R$ | Redundancy factor for critical data |
| $\lambda$ | Average latency of data collection |
| $E_{total}$ | Total energy consumption |
| $\tau$ | Average response time for high-priority data |

Where $E_j(t)$ is the energy level of the UAV $j$ at time $t$, $E_{min}$ is the UAV's minimum energy required threshold, $p_i(t)$ is the priority score of the packet, and $\theta$ is the threshold value that decides high-priority packets. If both conditions (1) and (2) are met, the UAV proceeds to collect $P_i(t)$ from the sensor or its assumed relay. This dynamic decision-making ensures that UAVs are dispatched only when necessary, preserving energy while guaranteeing that critical data is not delayed or lost due to weak BPL connectivity

3. Information processing and exchange: On reaching a sensor, UAVs retrieve $P_i(t)$ and for high-priority data $p_i(t) \geq \theta$), capturing additional information. If BPL capacity is insufficient, the UAV stores the data and retries transmission to a cloud when $C_{BPL}(t) \geq C_{min}$. Relay points also forward data to cloud storage via BPL, forming a closed-loop cycle. This phase optimizes real-time decision-making with efficient data routing, maximizing throughput while reducing latency and energy consumption.

## Problem statement

Consider a farm with $N$ sensor nodes and $M$ UAVs operating at a herding farm, measuring $200 \times 200 m^2$. Each sensor $i \in \{1, \ldots . N\}$ generates packet $P_i(t)$ at time $t$, transmitted via BPL to the nearest relay point. UAV $j \in \{1, \ldots . M\}$ retrieves $P_i(t)$ from a relay at distance $d_{ij}(t)$ with energy level $E_j(t) \geq E_{min}$. A priority function $p_i(t) \in [0, 1]$ assigns urgency, with $p_i(t) \geq \theta$) indicating priority data as determined in real time by sensor algorithms evaluating vital signs or environmental thresholds. The objective is to minimize UAV travel distance while maximizing timely data exchange, subject to energy and BPL capacity constraints.

**Coverage-Driven scenario (Scenario 1).** This scenario minimizes total UAV travel distance to ensure all sensors are covered, prioritizing comprehensive monitoring. The constraint equation is defined as:

$$\min \sum_{i=1}^{N} \sum_{j=1}^{M} (x_{ij}(t) \times d_{ij}(t)) + w. \lambda$$

(3)

Subject to:

$$E_j \geq E_{min} , \ \forall j \in \{1, \ldots \ldots . M\}$$

$$\sum_{j=1}^{M} x_{ij}(t) = 1, \ \forall \, i \in \{1, \ldots . . N\}$$

$$\sum_{j=1}^{M} x_{ij}(t) \geq R, \ \ \forall \, i \ with \ p_i(t) \geq \ \theta$$

$$C_{BPL}(t) \geq C_{min}$$

Where $x_{ij}(t) = 1$ if UAV $j$ collects data from the sensor $i$, else 0. $d_{ij}(t)$ is the distance from UAV $j$ to relay point that aggregates data from sensor $i$, $w$ is the weight for the timeliness penalty, $R$ is the redundancy factor that ensures critical data is collected by at least two UAVs for reliability, and $\lambda$ is the average latency of data collection across all sources. The algorithm for scenario 1 is presented as follows:

```
Algorithm 1. Coverage-Driven UAV-BPL synchronization
Input: N, M, Pᵢ(t), BPL relay locations, UAV range, BPL range, E_min, R
Output: UAV paths, data log, redundancy compliance
1. Initialize herding grid with BPL relay points
2. for each data source i=1 to N
3.         Generate Pᵢ(t), timestamp
4.         Transmit Pᵢ(t) to the nearest BPL relay point
5. end for
6. Assign relays to sensors by proximity
7. for each UAV j=1 to M do
8.         identify unvisited relay points within range
9.         while unvisited relay points remain do
10.           select the nearest unvisited relay k with min dᵢⱼ(t)
11.          if Eⱼ(t) ≥ E_min then
12.               Set xᵢⱼ(t) = 1 for all i at k
13.               move to k, record dᵢⱼ(t)
14.               collect all Pᵢ(t),mark critical data pᵢ(t) ≥ θ
15.               if pᵢ(t) ≥ θ and redundancy R>1 unmet then
16.                 signal another UAV to collect
17.               end if
18.              if C_BPL(t) ≥ C_min then
19.                 relay data to cloud (herder) via BPL
20.              else
21.                  wait
22.              end if
23.              mark k as visited
24.           end if
25.        end while
26. end for
27. Return to base when all relays have been visited
28. Compute λ = avg(t_collect(i) − t_gen(i)) for all i
29. Log λ for performance analysis
```

**Priority-Driven Exchange scenario (Scenario 2).** This scenario minimizes travel distance while prioritizing high-urgency data $p_i(t) \geq \theta$ and optimizing energy efficiency. The constraint equation is defined as:

$$\min \sum_{i=1}^{N} \sum_{j=1}^{M} (x_{ij}(t) \times d_{ij}(t)) + w_1 E_{total} + w_2.\tau$$

(4)

Subject to:

$$E_j(t) \geq E_{min} \; \forall j \in \{1, \ldots \ldots M\}$$

$$\sum_{j=1}^{M} x_{ij}(t) \geq 1, \forall \, i \; with \; p_i(t) \geq \theta$$

$$E_{total} \leq E_{max}$$

$$C_{BPL}(t) \geq C_{min}$$

Where $w_1$, $w_2$ are weights for energy and responsiveness. $\tau$ is the average response time for priority data. $E_{max}$ is the total energy budget. UAVs can visit multiple relays per time step if $E_j(t) \geq E_{min}$, with no explicit capacity constraint due to lightweight data packets. The algorithm for scenario 2 is presented as follows: The algorithm prioritizes high-urgency data, reassigning tasks if energy is low, with $\tau$ and $E_{total}$ computer for analysis.

```
Algorithm 2. Priority Driven UAV-BPL synchronization
Input: N, M, Pᵢ(t), pᵢ(t), BPL relay locations, UAV range, BPL range, Eₘᵢₙ, θ, Energy Consumption Rate
Output: UAV paths, priority data log, Eₜₒₜₐₗ₍ₜ₎, Tᵣₑₛₚ₍ₜ₎
1. Initialize farm grid with BPL relay points
2. for each data source i=1 to N
3.          Generate Pᵢ(t), timestamp
4.          Transmit Pᵢ(t) to the nearest BPL relay point
5. end for
6. for each UAV j=1 to M do
7.          Identify relay points within range
8.          for each relay k in range do
9.            retrieve Pᵢ(t), pᵢ(t),packets
10.            if pᵢ(t) ≥ θ and Eⱼ(t) ≥ Eₘᵢₙ then
11.              Set xᵢⱼ(t) = 1
12.              move to k at dᵢⱼ(t), record
13.              collect Pᵢ(t),capture imagery if needed
14.              if C_BPL(t) ≥ Cₘᵢₙ then
15.                 relay data to cloud via BPL
16.              else
17.             wait
18.               end if
19.            else
20.              Skip to the next packet/relay
21.            end if
22.          update Eⱼ(t) = Eⱼ(t) - (dᵢⱼ(t) × energy consumption rate)
23.          if Eⱼ(t) <Eₘᵢₙ:
24.                 reassign uncollected high-priority Pᵢ(t) to UAV with Eⱼ(t) ≥ Eₘᵢₙ
25.          end if
26.          compute τ = avg(114>>t_collect(i) - t_gen(i)) for high-priority i
27.          compute Eₜₒₜₐₗ = ∑ Eⱼ(t)
28. end for
```

## Simulation results and discussion

To evaluate the UAV-BPL synchronization framework, simulations were conducted for two scenarios: Coverage-Driven (Scenario 1) and Priority-Driven (Scenario 2). The setup considers a $200 \times 200 m^2$ dairy farm with 50 sensors, 5 UAVs, and 10 BPL relay points, using parameters from Section 3: communication range (100 m), energy threshold ($E_{min}$ = 20%), priority threshold ($\theta$ = 0.7), and redundancy factor (R = 2). Simulations ran 10 times with varying random seeds to ensure robustness, computing mean and standard deviation for key metrics: average latency ($\lambda$), response time for high-priority data ($\tau$), total energy consumption ($E_{total}$), coverage, and redundancy. The simulation parameters chosen intuitively used in this study are given in Table 4.

### Simulation results

**Scenario 1.** Coverage-Driven Exchange aims to ensure all sensors are visited, with redundancy for critical data $p_i(t) \geq \theta$. Fig 4 visualizes the results. (a) UAV paths, where blue circles represent sensors, red squares denote BPL relays, and black lines indicate UAV trajectories; (b) coverage and redundancy percentages; (c) data collection timeliness; (d) total distance and energy; (e) relay visitation frequency; and (f) energy per UAV. Over 10 iterations, Scenario 1

**Table 4. Simulation parameters.**

| S/N | Simulation parameters | Values |
|---|---|---|
| 1 | Herding size (m²) | 200×200 |
| 2 | Number of data sources | 50 |
| 3 | Number of UAVs | 5 |
| 4 | UAV Range (m) | 100 |
| 5 | Number of BPL relay points | 10 |
| 6 | UAV Energy rate (Energy consumption per 10 m) | 0.1 [39] |
| 7 | Redundancy factor for critical data | 2 |
| 8 | Minimum energy | 20 |
| 9 | Priority threshold | 0.7 |
| 10 | Max moves per UAV | 20 |

achieved a mean coverage of 99.2% (±2.5%) and redundancy compliance of 96.2% (±7.9%) for critical data. The average latency was 5.03 s (±0.89 s), and total energy consumption ($E_{total}$) was 7.9% (±1.1%).

The resulting UAV monitoring path plots for scenario 1 are shown in Figs 4–9. The plots reveal the distribution of data urgency around the herding farm, the effectiveness of UAV allocation to BPL relay locations, operational aspects such as energy management, relay visitation frequency, and distance optimization. The UAV paths in Fig 4 illustrate how the algorithms allocate UAVs to relays to maximize travel distance, covering all data sources. The coverage and energy metrics plot for Scenario 1's coverage, presented in Fig 5, shows 99.2% coverage of all 50 data sources, ensuring comprehensive herding-wide information exchange. Additionally, 96.2% critical data redundancy is achieved, meeting the redundancy goal to enhance reliability, albeit at a likely higher energy cost. Fig 6 shows Scenario 1's data collection timeliness plot, which shows the distribution of collection times for all 50 data sources, with an average latency of 5.03 seconds, indicating that most data is collected between 1 and 6 seconds, reflecting the algorithm's focus on comprehensive coverage but with varied delays due to exhaustive relay visits. Scenario 1's distance plot highlights the high resource demand of its exhaustive approach to achieving 99.2% data coverage by visiting all BPL relay points. Fig 7 shows the total distance and energy for the scenario. The algorithm's aim to cover all 10 BPL relay points for 100% data collection from 50 sources is reflected in Scenario 1's relay visitation frequency plot presented in Fig 8, which indicates that relays 1–5, and 7, 9–10 were visited twice each, while relays 6 and 8 were visited once. Higher visits to specific relays are probably caused by the need for redundancy for critical data. As Scenario 1 thoroughly covers all 50 data sources via 10 BPL relay points, UAV 1 probably handles more relays to achieve higher coverage, as evidenced by the energy consumption per UAV bar chart in Fig 9, which shows that UAV 1 used more of its energy in comparison to the others.

**Scenario 2.** Priority-Driven Exchange prioritizes high-priority data $p_i(t) \geq \theta$. The results for this scenario are given in Figs 10–15. Fig 10 presents the UAV paths, with sensor colors indicating priority scores. Fig 11 shows priority coverage and energy. Similarly, Figs 12 and 13 show the response time distribution and total distance covered, respectively, while Figs 14 and 15 show relay visits and cumulative coverage. The mean priority data coverage was 100% (±0.0%), with a response time (τ) of 4.94 s (±0.18 s) and $E_{total}$ of 11.8% (±4.8%).

The resulting UAV monitoring path plots for scenario 2 are shown in Fig 5. The paths illustrate how the algorithms allocate UAVs to relays to maximize travel distance, prioritizing either high-urgency data. Scenario 2's coverage and energy metrics plot reveals 100% coverage of high-priority data, reflecting its focus on urgent information like health alerts while satisfying energy constraints ($E_{min} = 20\%$) and a low ~12% total energy usage, highlighting its energy efficiency but trading off full coverage for selective responsiveness in herding operations. Scenario 2's high-priority response time plot illustrates the response times for high-priority data with a mean 4.94 seconds, and most of the

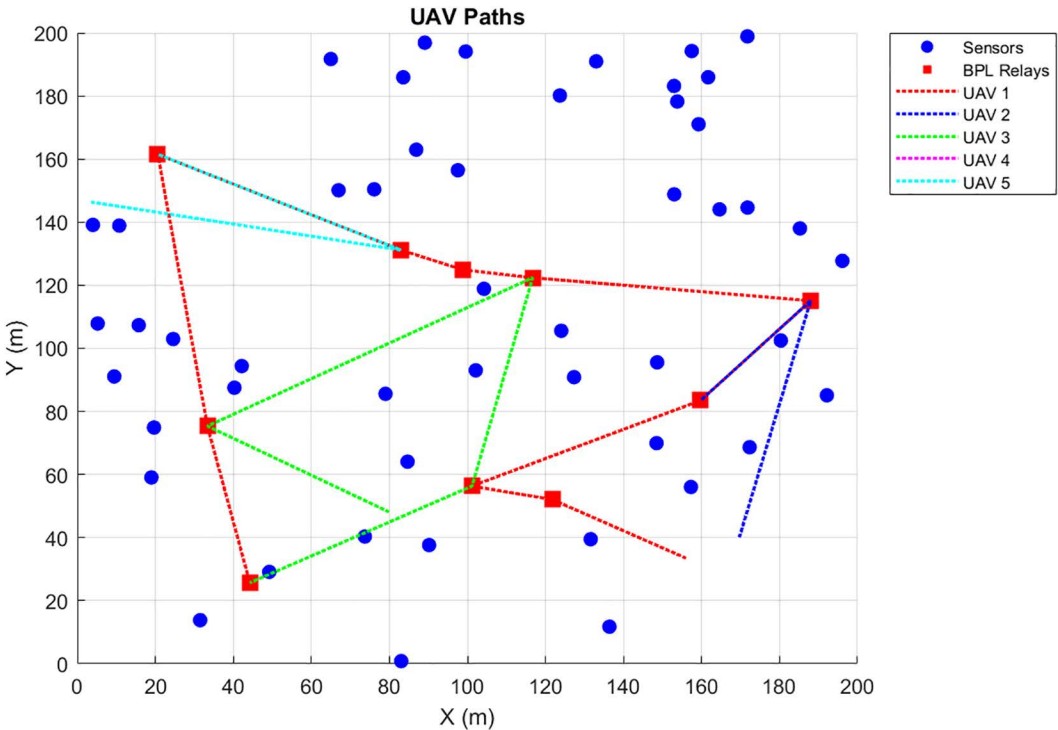

**Fig 4. UAV paths for Coverage-Driven scenario (blue circles: sensors, red squares: relays, black lines: UAV trajectories).**

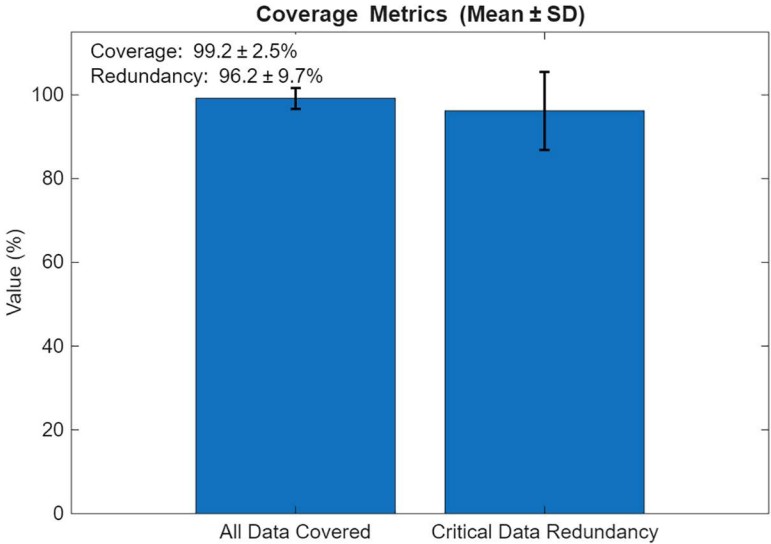

**Fig 5. Coverage and redundancy metrics for Coverage-Driven scenario.**

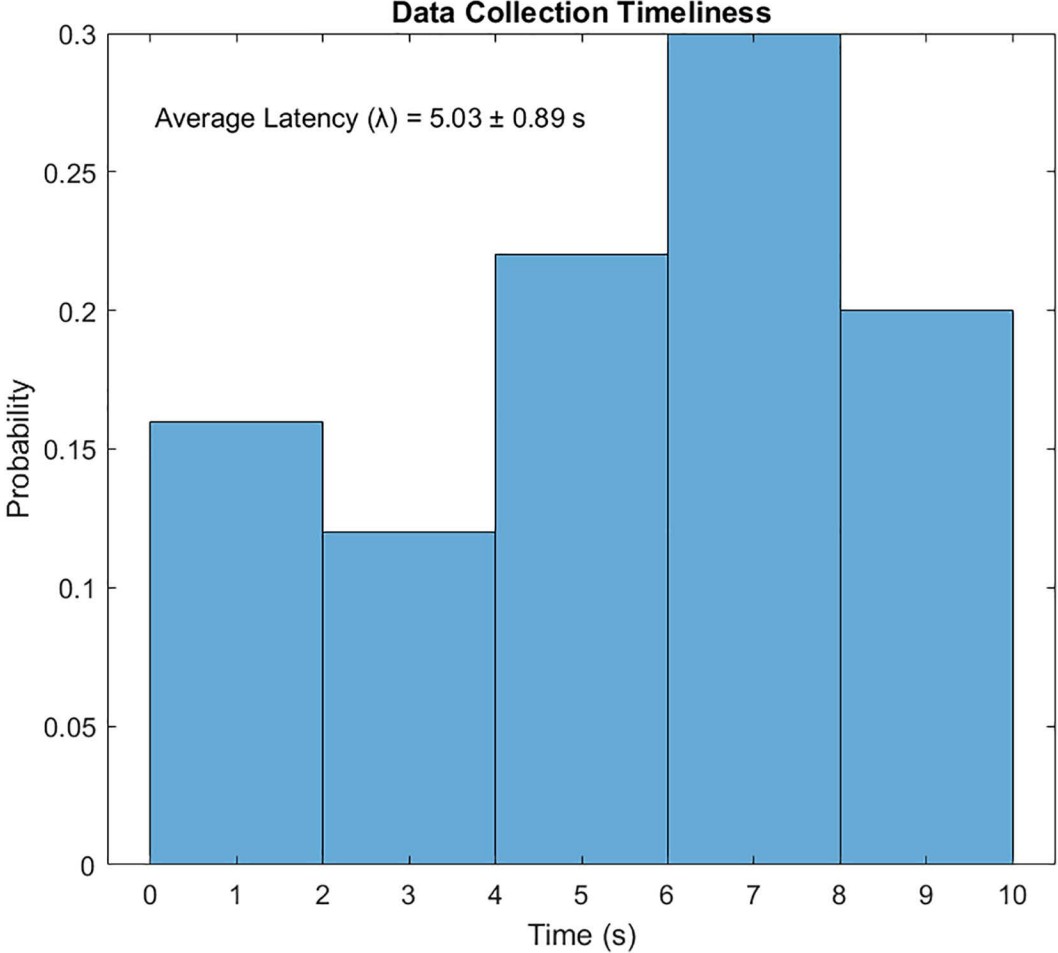

**Fig 6. Timeliness distribution for Coverage-Driven scenario.**

time clustering between 2 and 8 seconds, demonstrating faster responses to urgent data, such as health alerts, due to its selective approach. The total UAV distance plot for Scenario 2, on the other hand, shows a much higher total distance of roughly 2000m. Despite its priority-driven approach focusing on high-urgency data, this suggests more travel possibly due to the reassignment of uncollected high-priority data to available UAVs, though it maintains energy efficiency that was previously shown as ~12% total usage by selectively targeting urgent relays, balancing responsiveness and resource conservation in herding operations. Scenario 2's relay visitation frequency plot shows that some relays were visited three times each, while some others were not visited. Scenario 2's cumulative coverage over sources line plot, on the other hand, shows that coverage increases stepwise to as data sources are processed. This is consistent with the priority-driven focus on high-urgency data. The results of the simulation's scenario is presented in Table 5.

Conclusively, it can be stated that Scenario 1 is suited for herding operations prioritizing reliability and complete data collection such as routine monitoring of livestock and environmental conditions while Scenario 2 is ideal for time-sensitive applications such as rapid detection of health issues or environmental hazards. Therefore, the choice depends on whether the herding operations values exhaustive coverage or rapid response to critical events.

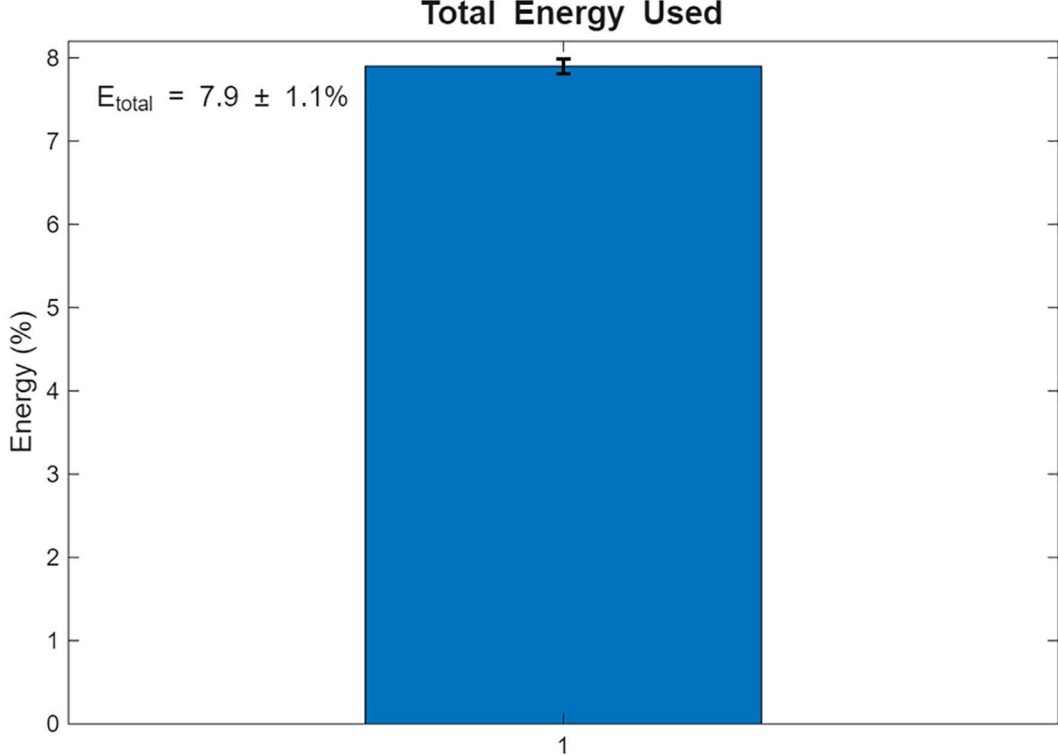

**Fig 7. Total distance and energy for Coverage-Driven scenario.**

## Discussion

Redundancy Impact: In Scenario 1, the redundancy factor (R = 2) ensures critical data is collected by at least two UAVs, achieving 95.6% compliance. Thus, this indicates that the latency increases with low redundancy. On the other hand, the total energy $E_{total}$ increase due to additional UAV flights, as shown in Fig 4(e). However, this enhances reliability for critical data such as health alerts which is crucial for herding farm operations.

Response Time Significance: Scenario 2's τ = 4.94s ensures rapid collection of high-priority data, enabling timely responses to livestock health or environmental alerts. This supports early decision making such as dispatching veterinarians for urgent cases. Scenario 2's priority coverage uses BPL's high bandwidth [38] and UAV mobility [40] to overcome limitations of low-bandwidth wireless systems.

Parameter Sensitivity: Simulations varying farm size (100 × 100–400 × 400 m²), UAVs (3–10), and sensors (25–100) show that larger farms increase latency and total energy due to longer UAV paths. Increasing UAVs reduces latency. More sensors improve coverage but slightly increase average response time in Scenario 2, indicating scalability trade-offs. The framework's robustness, low latency, and energy efficiency make it a viable solution for smart herding operations. Thus, capable of addressing gaps in existing methods while balancing reliability and real-time performance.

## Conclusions

This study utilized the farm's existing electrical grid via Broadband over Power Line (BPL) communication to enable cost-effective, scalable data collection and communication. Simulations used across a 200 × 200m² farm with 50 sensors,

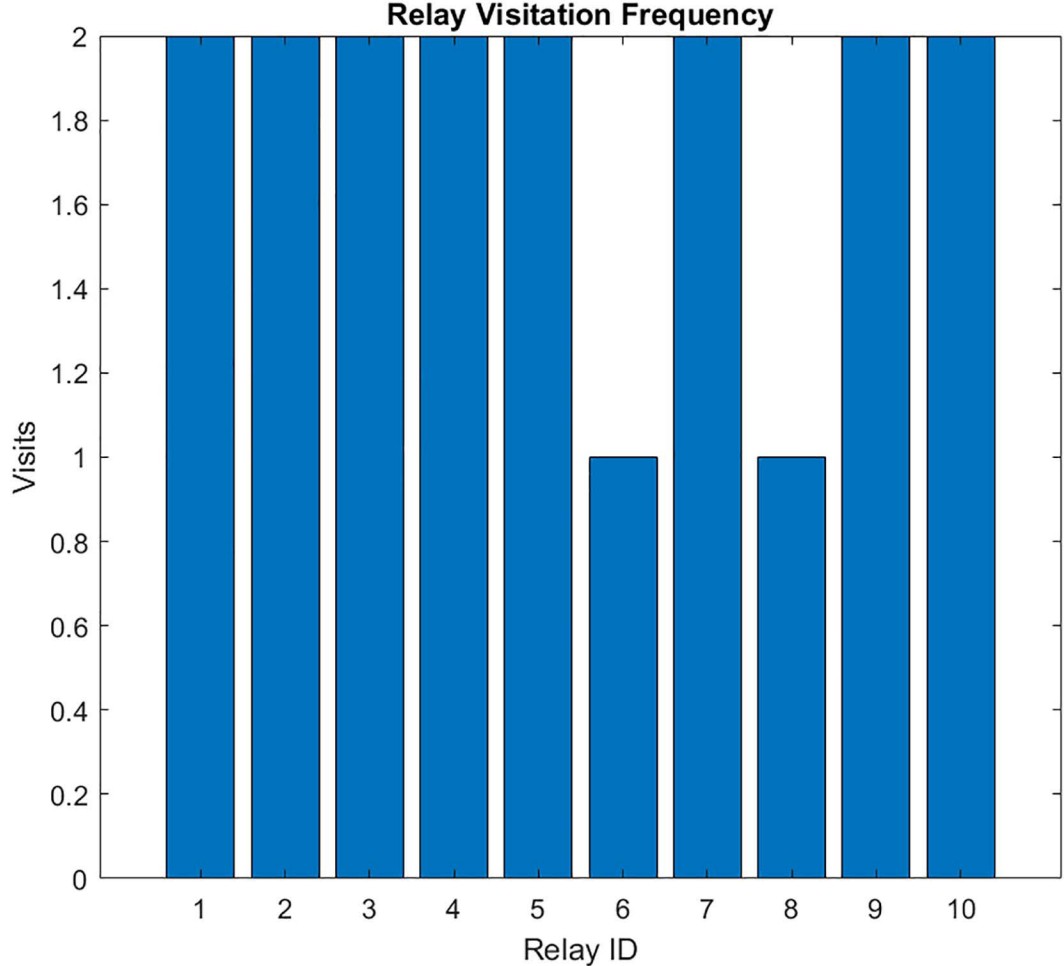

**Fig 8. Relay visits for Coverage-Driven scenario.**

5 UAVs, and 10 BPL relays demonstrate its efficacy. In Scenario 1 (Coverage-Driven), the framework achieves 99.2% coverage and 96.2% redundancy for critical data, with an average latency of 5.03s and total energy consumption ($E_{total}$) of 7.9%. Scenario 2 (Priority-Driven) delivers a response time of 4.94s for high-priority data, with 100% coverage and $E_{total}$ of 11.8%. These results will enable timely interventions, such as dispatching veterinarians for livestock health alerts and adjusting environmental controls. Thereby, improving farm efficiency and animal welfare. The framework's novelty lies in synergizing BPL's high-bandwidth, reliable infrastructure (IEEE 1901) with UAVs' mobility. Unlike low-bandwidth wireless systems, it ensures robust data transfer even under high sensor loads. However, limitations include dependency on stable BPL infrastructure, which may face noise or outages, and scalability challenges for very large farms hectares. These results provide herders with a flexible framework to optimize farm management by customizing the algorithm to meet specific requirements, such as energy-efficient emergency response or thorough monitoring. The system is an effective instrument for raising agricultural productivity due to its scalability, dependability, and flexibility in responding to changing herding conditions. Future work includes real-world deployments to validate performance under varying conditions, developing adaptive synchronization algorithms to handle dynamic sensor priorities, and integrating IoT analytics for predictive farming.

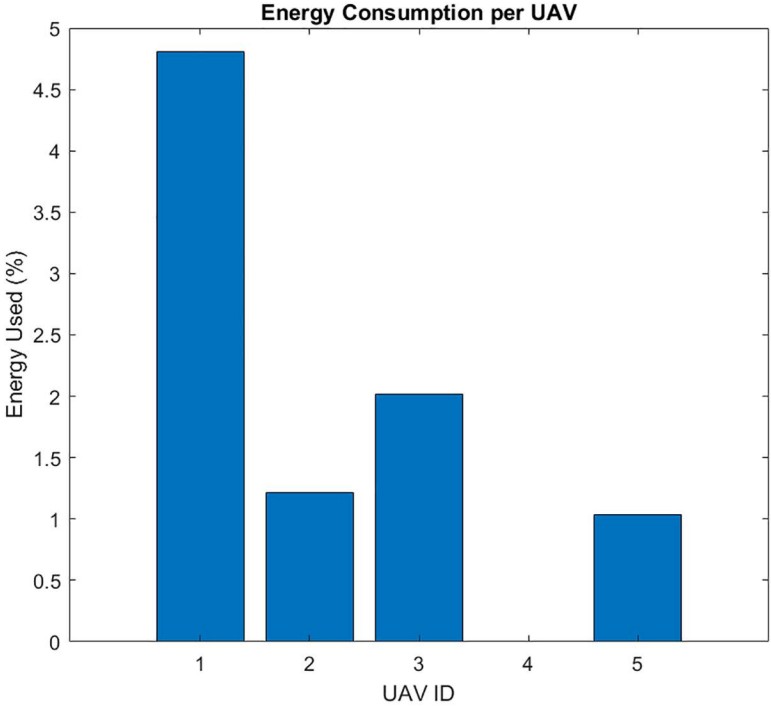

**Fig 9. Energy per UAV for Coverage-Driven scenario.**

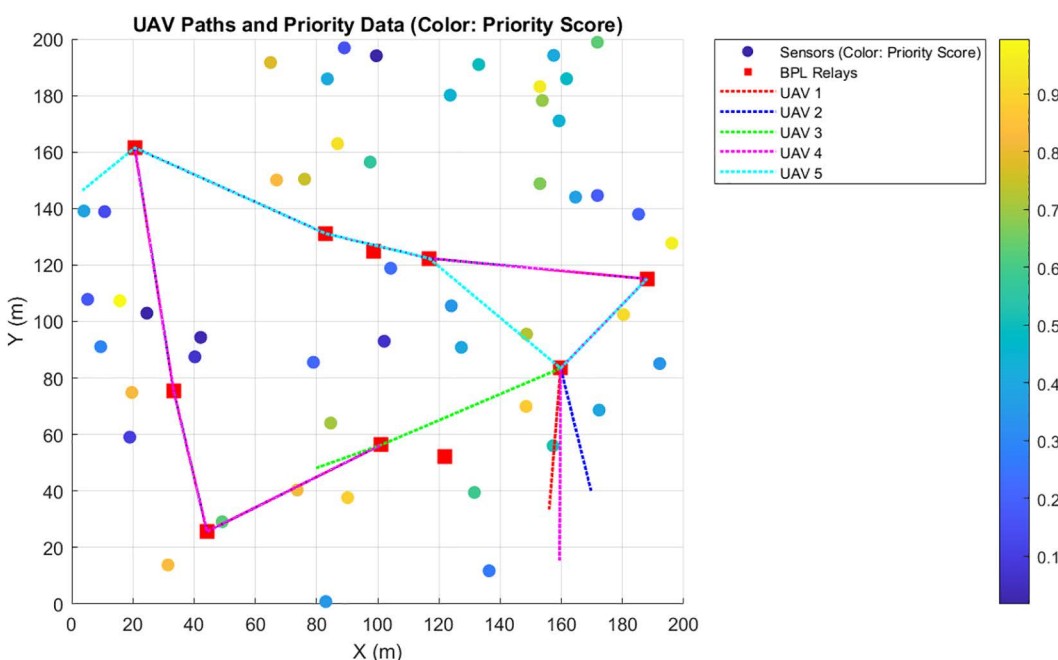

**Fig 10. UAV paths for the Priority-Driven scenario (sensor colors indicate priority, yellow high to blue low).**

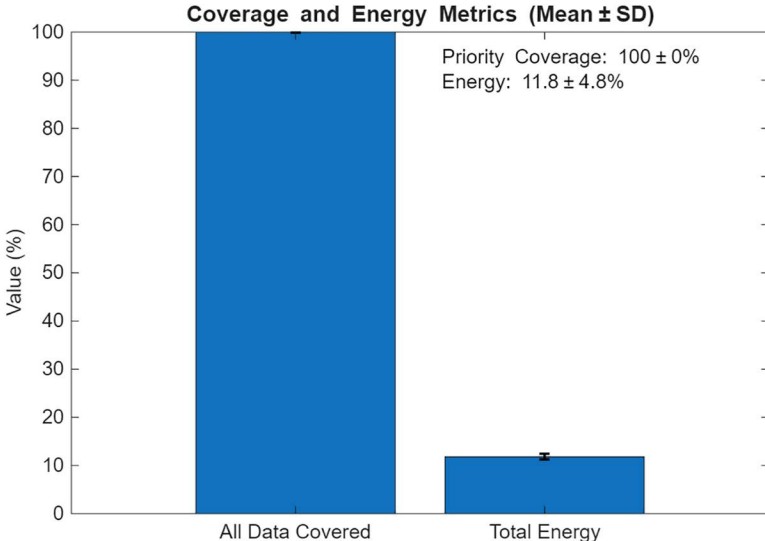

**Fig 11. Priority coverage and energy for the Priority-Driven scenario.**

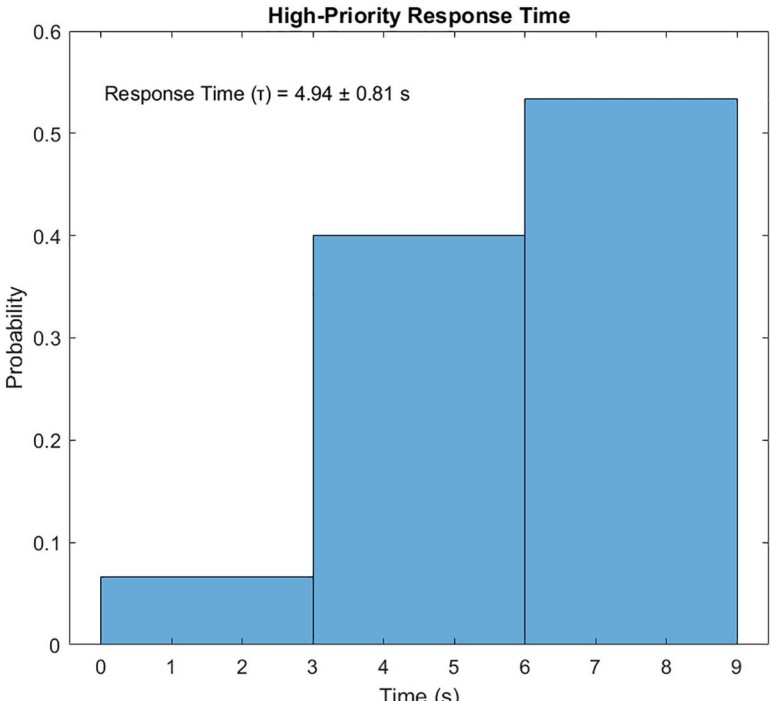

**Fig 12. Response time distribution for the Priority-Driven scenario.**

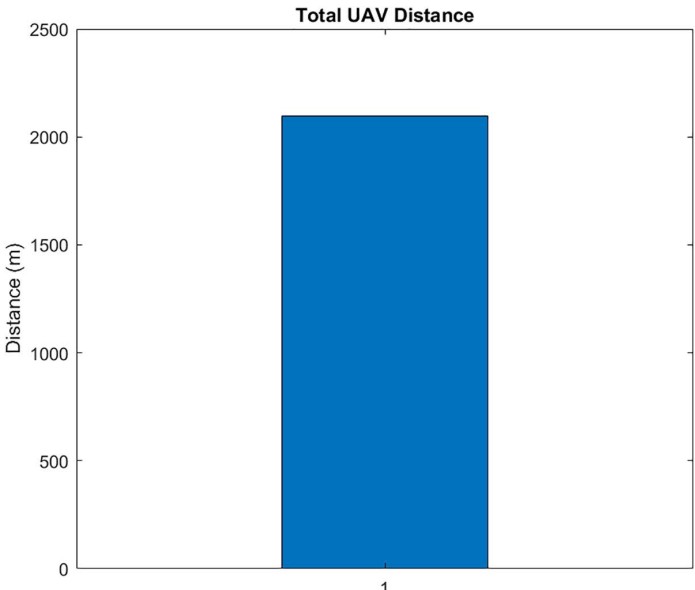

**Fig 13. Total distance for the Priority-Driven scenario.**

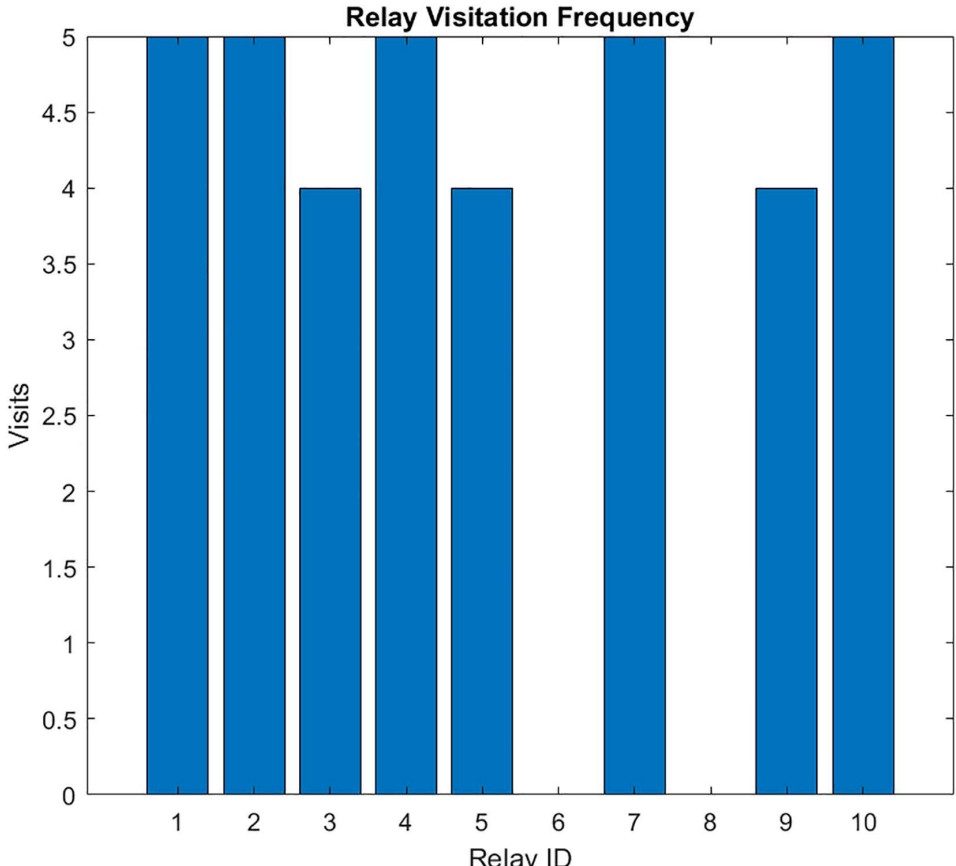

**Fig 14. Relay visits for the Priority-Driven scenario.**

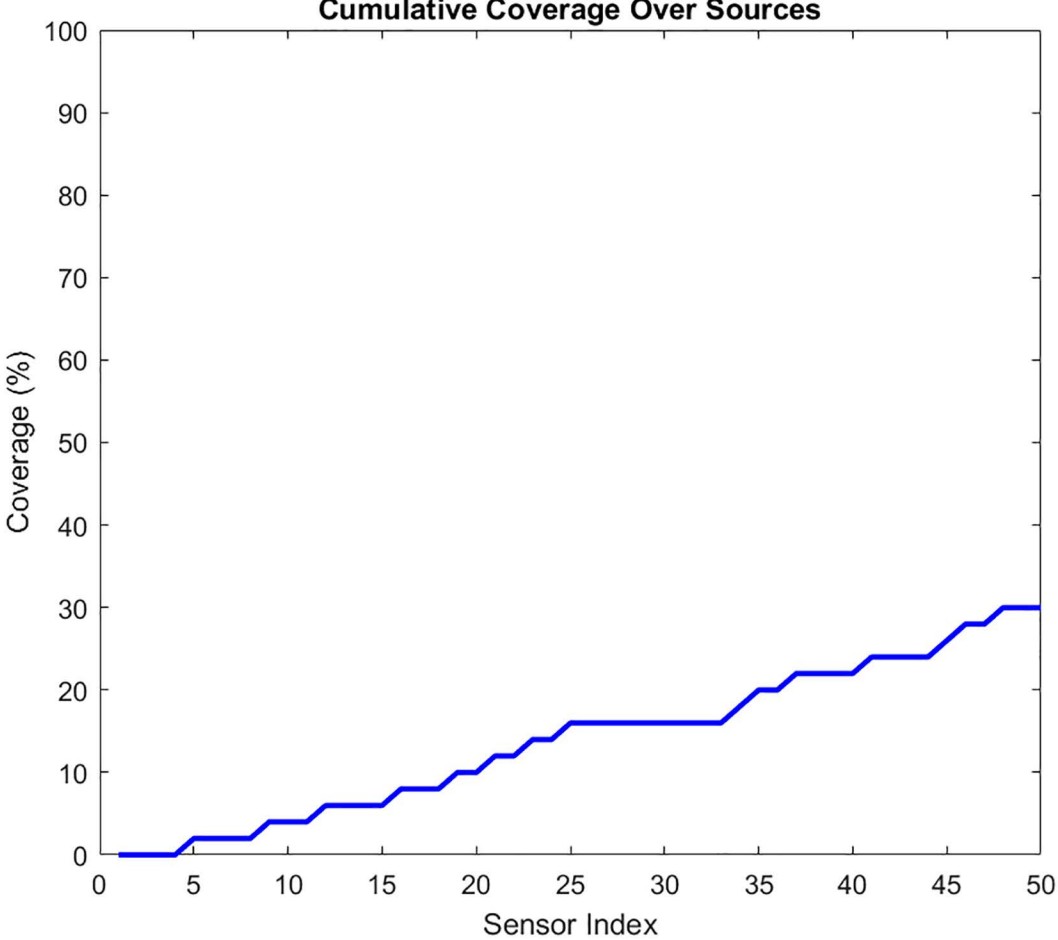

**Fig 15. Cumulative coverage for the Priority-Driven scenario.**

**Table 5. Simulation results for Scenarios 1 and 2.**

| Metric | Scenario 1 | Scenario 2 |
|---|---|---|
| Latency/Response time | 5.03s | 4.94s |
| Energy Consumption | 7.9% | 11.8% |
| Coverage | 99.2% | 100% |
| Redundancy Compliance | 96.2% | ------- |

## Author contributions

**Conceptualization:** Bashir Olaniyi Sadiq, Motirh Al-Mutairi, Yusuf Abubakar Sha'aban.

**Data curation:** Houssem Rafik El-Hana Bouchekara, Khalid Almutairi.

**Formal analysis:** Yusuf Abubakar Sha'aban.

**Funding acquisition:** Motirh Al-Mutairi.

**Methodology:** Bashir Olaniyi Sadiq, Yusuf Abubakar Sha'aban, Houssem Rafik El-Hana Bouchekara.

**Project administration:** Yusuf Abubakar Sha'aban, Houssem Rafik El-Hana Bouchekara.

**Resources:** Motirh Al-Mutairi, Houssem Rafik El-Hana Bouchekara, Khalid Almutairi.

**Software:** Bashir Olaniyi Sadiq.

**Supervision:** Yusuf Abubakar Sha'aban, Houssem Rafik El-Hana Bouchekara.

**Validation:** Motirh Al-Mutairi, Khalid Almutairi.

**Visualization:** Khalid Almutairi.

**Writing – original draft:** Bashir Olaniyi Sadiq, Yusuf Abubakar Sha'aban.

**Writing – review & editing:** Motirh Al-Mutairi, Yusuf Abubakar Sha'aban, Houssem Rafik El-Hana Bouchekara, Khalid Almutairi.

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
