## [Decision Letter · Decision Letter 0]

30 Jul 2025

Dear Dr. Sha'aban,

Thank you for submitting your manuscript to PLOS ONE. After careful consideration, we feel that it has merit but does not fully meet PLOS ONE’s publication criteria as it currently stands. Therefore, we invite you to submit a revised version of the manuscript that addresses the points raised during the review process.

Dear authors,

I appreciate your effort in this submission. However, you are required to make thorough improvements as per the reviewer comments, especially regarding the realistic simulation settings and the use of appropriate operational parameters. Please note that some reviewers might have recommended some citations in their review. I recommend that you carefully evaluate these references and determine whether they are relevant to your current study. Feel free to disregard any references that do not align with the content of your manuscript, this will not affect the acceptance or rejection of your paper.

We look forward to receiving your revised manuscript.

Kind regards,

Antar S. H. Abdul-Qawy, Ph.D.

Academic Editor

PLOS ONE

Journal Requirements:

Princess Nourah bint Abdulrahman University Researchers Supporting Project number (PNURSP2025R241), Princess Nourah bint Abdulrahman University, Riyadh, Saudi Arabia

6. Please upload a copy of Figure 3, to which you refer in your text on page 7, 12, and 13. If the figure is no longer to be included as part of the submission please remove all reference to it within the text.

7. Please include a copy of Table 2 which you refer to in your text on page 5 and 10.

Reviewers' comments:

Reviewer's Responses to Questions

**Comments to the Author**

1. Is the manuscript technically sound, and do the data support the conclusions?

Reviewer #1: Yes

Reviewer #2: Yes

Reviewer #3: Yes

Reviewer #4: Yes

Reviewer #5: Partly

Reviewer #6: No

Reviewer #7: No

2. Has the statistical analysis been performed appropriately and rigorously?

Reviewer #1: N/A

Reviewer #2: N/A

Reviewer #3: Yes

Reviewer #4: Yes

Reviewer #5: N/A

Reviewer #6: No

Reviewer #7: No

3. Have the authors made all data underlying the findings in their manuscript fully available?

Reviewer #1: No

Reviewer #2: No

Reviewer #3: Yes

Reviewer #4: Yes

Reviewer #5: No

Reviewer #6: Yes

Reviewer #7: No

4. Is the manuscript presented in an intelligible fashion and written in standard English?

Reviewer #1: Yes

Reviewer #2: Yes

Reviewer #3: Yes

Reviewer #4: Yes

Reviewer #5: Yes

Reviewer #6: Yes

Reviewer #7: No

Reviewer #1: The paper presents an innovative and practical framework that leverages the complementary strengths of UAVs and Broadband over Power Line (BPL) technology to address the communication challenges in dairy farming operations, particularly in rural areas with limited wireless coverage. Its core contribution lies in the novel "Sky and Wire" synchronization algorithm, which intelligently integrates aerial and wired infrastructure to optimize data collection, balance energy efficiency, and ensure timely relay of both routine and high-priority information. However, several points need clarification :

- Improve the quality of figures.

- What are the specific assumptions about the farm electrical infrastructure (e.g., BPL availability, grid reliability, noise/interference levels)?

- How is the synchronization between UAV and BPL established and maintained in practice?

- Are there limitations on the number of UAVs or relay nodes? How scalable is the system for larger farms?

- What factors were considered when deciding the energy consumption model, and does it account for real UAV dynamics (e.g., wind, payload)?

- How does the system prioritize data urgency ( ≥ 0.7)? What is the mechanism for determining urgency levels in real time?

- What happens if BPL communication is disrupted or the grid is unstable, are there fallback mechanisms?

- How does the proposed framework contribute to sustainable agriculture, especially in terms of reducing manual labor, improving animal welfare, or optimizing resource use?

Reviewer #2: Abstract is very poor and needs to convey the proposed idea or solution in a unique way.

Need to rewrite Introduction properly It’s a vague introduction Meaningless

Somewhat good but not a thorough analysis. Not able to portray research gaps

Methodology is a mediocre

Results are average but a bit promising

Conclusion needs to be revised and fine-tuned

Adequate but not that much relevant

In addition, it has been observed that the authors do not demonstrate clearly and objectively the relevance of this research. When reading the research, we observed that the authors describe it subjectively, that is, the importance of the respective results, methodologies, and consequently of this article is not evident.include recent references 2023/24

Reviewer #3: The authors focused on SKY AND WIRE: A UAV-BPL SYNCHRONIZATION ALGORITHM FOR INFORMATION EXCHANGE IN HERDING OPERATIONS. Please address the following suggestions.

- The context and problem statement in the introduction are missing. It should be more elaborate.

- What is the focus and motivation of the work? These should be added in the introduction.

- Improve by emphasizing the key novelties and contributions, incorporating essential numerical results. Ensure conciseness and focus.

- Clearly highlight research gaps, novelties, and contributions.

- Added the main contribution at the last of the introduction.

- The manuscript (Introduction) has deficient citations to very important works published before; you can add these works:

https://doi.org/10.1016/j.iot.2025.101704; https://doi.org/10.1016/j.sciaf.2024.e02527; https://doi.org/10.1016/j.atech.2025.100979; https://ieeexplore.ieee.org/document/10548972 ; https://link.springer.com/chapter/10.1007/978-981-96-3094-3_10

- Rewrite the description of the pseudocode of the system in a professional.

- The Section of the Results and Discussion needs to add a more detailed description of the results presented in the figures and Tables should be provided, incorporating both qualitative and quantitative analysis.

- Detail the approach used to validate numerical results.

- Please improve the conclusion section and add the Future Work.

Reviewer #4: Observation 1:

BPL integration with UAV is a promising concept to which I agree.

Normally the transmission lines will be high power sources and directly connecting

them with the IoT enabled fog devices for data extraction and forwarding could be dangerous.

Is there a safe way to conduct this? I mean using step down transformers or some solution.

Using UAVs to extract data and BPL points for forwarding and extracting must be studied

from the perspective that what impact it will have on low power devices and at what

level human interaction in this environment will take place as high voltage sources are

involved. Therefore the authors should address the safety and hazards along with this

coupled scenario.

Observation 2:

Table 2 has formating issues please re format it.

Observation 3:

In the coverage driven scenario 1 explain what eqation 3 means clearly.

The optimization equation must be linked and explained in light of equation 3.

In the model what is the role for the redundancy. Is this something that you need meaning

that redundancy is required or this is some penalty value meaning that it is an over head. THis

needs to be cleared.

Observation 4:

Table is written again on the alogrithm 1 of the coverage driven scenario.

Observation 5:

Line 22 of the alogirthm 1 computes the average. What will you do with this and where

is it used again in the algorithm. is this value used in another algorithm. please

clearly mention it either before the algorithm or after it.

Observation 6:

The results of the study are good and are showing what the authors intends to achieve but

the resutls should be clearly explained so that the reader is able extract what the lines means.

For example in figure 3 in scenario 1 and scenario 2 what is the meaning of black lines that

are connected to the data sources and the BPL relays. What does this black means in both the scenarios

there is no explanation related to that.

Reviewer #5: 1) Based on the abstract, the research problem and research gaps have not been described. It is therefore difficult to discern what really motivated the development of the proposed framework.

2) This paper used performance indicators such as coverage, energy usage, frequency of relay visits, and response timeliness using MATLAB simulations. What were the numerical values obtained for these indicators and what do these obtained values imply?

3) The main contributions of this paper reflects marginal novelty. You need to restructure this section so that the acclaimed novelty can be made clearer.

4) After Table 2, add a paragraph to describe a summary of the identified research gaps. This should be followed by a clear explanation on how the proposed framework helps alleviate these gaps.

5) It is not clear why you are referring Algorithm 1 as Table 2. Similarly, it is not clear why you are referring Algorithm 2 as Table 3. You already have Table 2 which gives a Summary of Relevant Works under Section 2.

6) Ensure that all the algorithms are well discussed within text.

7) Consider adding a table of all symbols and notations used throughout this paper.

8) In Table 4, you have given the Simulation Parameters. However, you have failed to describe the simulation tool/environment under which these parameters are deployed.

9) What informed the selection of the simulation parameters in Table 4? How will the results be affected if different values are deployed?

10) For each presented tables/graphs in section 4, given elaborate descriptions as well as interpretations.

11) In the conclusion section, describe some of the limitations of the proposed framework. Thereafter, explain how these limitations may be addressed.

Reviewer #6: The authors present "Sky and Wire", a synchronization algorithm that combines Unmanned Aerial Vehicles with Broadband over Power Line (UAV-BPL) for herding operations. Two scenarios are presented: Scenario 1, the coverage-driven scenario, and Scenario 2, the energy-efficient priority-driven exchange scenario. The algorithms of these two scenarios are validated throughout MATLAB simulations.

Comments to the authors are divided into sections or subsections of the document.

Abstract:

The abstract needs to be revisited. In the section where the two scenarios are described, the name of the first scenario differs from the one stated in the problem statement section. It's unclear where the description of scenario 2 begins, and the name of this scenario is not given in the abstract.

Introduction:

It is unclear what the advantage of using drones in this proposed algorithm is, and why the data from all sources is not sent directly to the farm operators or the cloud throughout BPL. The Introduction does not really set the work in context.

Related work:

Several abbreviations without descriptions are used. There are some typos, for example, right before the table, ofsoem (of some?).

Methodology:

The quality of the Figures should be improved, especially Fig. 3 (in this section, because Fig. 3 is repeated in the simulation and results section); there are blue boxes marked as “Repeater” instead of "relay points" as throughout the document. The fonts in the image have different sizes. In the document, "the cloud" is employed instead of "internet", as in the image. "LV line", "CPE", and "Head-End" are never defined.

Suffixes "_ij" on "d_{ij}" are not defined, nor "Ø", "E_min", or "C_min".

In equation (2), there is an extra ")" at the end. Right after this equation, E_i should be E_j.

Problem statement:

In the proposed algorithm, the authors don't take into account the flight time consumption in the data transportation between relay points and UAVs. It should be mentioned why it is neglected. What are the communication protocols involved in the data collection?

The names of both scenarios differ throughout the document.

Suffixes "_ij" on "x_{ij}" are not defined.

In equations (3) and (8), the operator between x_{ij} and d_{ij} looks like a cross product, clarify.

In the paragraph after equation (7), x_{ij} and d_{ij} have opposite definitions to the previous definitions.

Simulation Results and Discussion:

Minimum energy is the total percentage or the used percentage? (Table 4). The initial conditions for UAVs are not given, nor is the starting point; if they are set randomly, this must be specified and justified.

After Table 4, in the discussion of Fig. 3 of this section, relay spots and UAV initial positions are both described as red squares, which makes it impossible to understand.

Then, it is said that the paths illustrate how the algorithms allocate UAVs to relays to maximize travel distance. This means that with less energy consumption, more distance can be covered?. Explain in more detail.

Fig. 3 is difficult to understand. For example, on Table 4, it is said that there are 10 BPL relay points; however, only 7 can be seen in each scenario, and the UAVs' initial positions are not shown. UAV paths are not dashed lines as described, and the path for each UAV can't be distinguished.

The paragraph after Fig. 3 states that in scenario 2, 100% of high-priority data is covered, whereas in the abstract, it's stated that only 80% is covered. What is the reason for this difference?

The description of Fig. 4 states that Scenario 2 accounts for approximately 20% of total energy usage; however, it can be seen from Figure 4 that this is far from the 20% mark, closer to 10%. Maybe this kind of data could be better presented in a table with the exact value. Since it's a simulation, data should be accessed accurately.

After Fig. 4, is it timelines or timeliness? Same on the title of Fig. 5.

typos: "secs" "moves".

The 20 movements per UAV are not taken into account in the algorithms described above. Explain why this is required and how it is taken into account in the algorithms.

Units on Fig. 5.

Discussion about Fig. 6 states that the resource demand is highlighted; however, due to the scale, it's difficult to see the percentage of energy used, and it's not provided in the discussion. A Table with the actual value might be better.

A deeper analysis is expected in the discussion of the results presented in Fig. 7 for Scenario 1. Why are there three relays with higher visitation frequency?

The bar graph of Scenario 1, Fig. 8, doesn't coincide with its description. The vertical scale in Fig. 8 is wrong.

In general, as different data are presented for each scenario, it's difficult to compare them; similar data for both scenarios could say more about the main differences and advantages for a given need.

Conclusion

The BPL dependability highlights the framework; however, the effect of the increased complexity of the wireless part of the framework is unclear. Limitations of the approach should also be mentioned. More simulations should be made, giving variations, for example, on the distributions of relay spots or the distribution of urgency, and the initial conditions of the drones, to arrive at the conclusions.

The code is not available on the given link.

Reviewer #7: This paper proposes a synchronization algorithm for dairy farm management that integrates UAVs with BPL to optimize data collection, either by maximizing coverage or by prioritizing urgent data and energy efficiency. I appreciate the authors efforts and the interesting topic addressed. However, the paper lacks valid simulation settings and in-depth discussion and detail in several areas that need to be further explored. There are also a number of weaknesses that require serious and careful revision. Below are some of my observations and comments, which should be thoroughly addressed and incorporated into the manuscript, not just provided as responses.

1. The abstract needs improvement in terms of language, flow, and clarity. Please revisit your abstract and rewrite it professionally and concisely, clearly highlight your objectives, contributions, and methodology.

2. You mentioned two scenarios in the abstract, but only one is described, while the other is unclear and appears mixed into the description.

3. The entire paper needs thorough English proofreading, as there are several unclear statements, improper flow, and ambiguities throughout the text.

4. Apart from satellite communication, the paper requires more statistical analysis regarding the cost-effectiveness and affordability of using UAVs compared to traditional wireless monitoring methods.

5. Please correct “ ” in “Where ( ) is the energy level” to “ j” on page 8.

6. Please remove the repetition in the fourth paragraph of Section 3.1, starting with “The aim is to minimize the total distance UAVs travel to…”

7. Please revise the section on “The Coverage-Driven Scenario (Scenario 1),” its equations, and the algorithm, as there are some contradictions between your descriptions, equations, and algorithm. For example, there is inconsistency regarding whether UAVs collect data from sensors or relay points.

8. It is not clear how BPLs are used, what standards are followed, and how they operate. Please elaborate and provide technical details on BPL operation and its integration with UAVs.

9. There should be a separate section explaining the “UAV-BPL synchronization” in detail, including how it is achieved and what are the transmission protocols used from sensors to BPLs, through BPLs to its relay points, and from relay points to UAVs.

10. What are relay points made of? Please clarify.

11. There seems to be a contradiction between your objective equation no. 3 and the description of xij and dij . Based on equation 4, I assume “xij(t) is the binary decision variable for UAV movement (xij(t) =1 if UAV j collects data from source i, 0 otherwise),” and “dij(t) is the distance from UAV j to the BPL relay serving source i” correct? This is different from your description. Please revisit all your equations and descriptions and clarify the relationship between variables for the reader.

12. Also, in Algorithm 1, there are contradictions: “ ( )” is used as the distance to the nearest relay, while xij(t) is used as a decision variable. Please ensure consistency.

13. I did not see a check for CBPL(t)≥Cmin before relaying data, as indicated in line 17 of the algorithm 1.

14. It is unclear what is considered a data source. In Section 3, you mention data sources 1 to N (sensors) creating packets and sending them to relay points, but later in the scenario description, it seems you refer to relay points as data sources visited by UAVs. Please be consistent.

15. I did not see any UAV energy/range check in Algorithm 1. Is it used as input? Why?

16. I suggest using a threshold symbol other than ∅, which means “empty set”

17. In Algorithm 2, the energy consumption rate is used, but how is it determined? Please explain or provide a proper citation.

18. Please remove the two captions “Table 2 Coverage-Driven” and “Table 3 Energy-Efficient Priority-Driven Exchange.” You can refer to them as Algorithm 1 and 2, which are already captioned.

19. How do you define "critical" data sources, or in what cases we know that it is critical? and how is R chosen?

20. Does your model allow a UAVs to visit more than one data source per time step? If so, should there be a constraint on UAVs capacity or energy?

21. In your simulation, you mention a 200x200 area. Is this in meters? If so, this is a small-scale farm that could be covered by traditional wireless methods, while your solution aims to address large-scale farms. How do you validate that your solution meets your design objectives?

22. This simulation is conducted for a single iteration of target monitoring, which definitely results in higher performance metrics and very low energy consumption. For validity, the simulation should be extended to cover a specified duration that emulates several hours or days of operation.

23. Your simulation lacks context and a valid environment. Please mention your simulation environment (tools, libraries), and explain how you simulate UAVs and BPLs, the standards and protocols assumed for communication and transmission, and the integration of UAVs and BPLs. Even though you have “Communication Technologies Comparison” in you related work section, but these were absent in you simulation part.

24. The simulation also lacks in-depth analysis, comparison, and justification. It appears that results are based on set parameters without reference points. Please provide more analysis and justification.

25. In your simulation results, you claim “100% coverage of all 50 data sources” and “highlighting its energy efficiency”. This result is expected given your design, implementation, and parameter selection, but there is no comparison with other solutions to demonstrate novelty. Please provide comparisons with some existing approaches and highlight where your solution is superior.

26. What energy consumption model did you use?

27. How did you choose the parameters in Table 4 (UAV range, energy rate, redundancy factor, critical data, etc.)? Are they based on datasheets, chosen arbitrarily, or taken from references?

28. In Figure 1, the legend does not reflect what is shown in the figure. For example, all UAVs 1 to 5 have the same line style and color, so why you listed them separately? Also, the colors (green, blue-green, yellow, etc.) given on the scale 0.2 to 0.8 are not explained.

29. I also suggest diversifying your results presentation by including tables along with graphs.

30. Please use a standard referencing style according to the journal guidelines.

**Do you want your identity to be public for this peer review?** For information about this choice, including consent withdrawal, please see our Privacy Policy

Reviewer #1: No

Reviewer #2: No

Reviewer #3: No

Reviewer #4: No

Reviewer #5: No

Reviewer #6: No

Reviewer #7: No

---

## [Decision Letter · Decision Letter 1]

8 Sep 2025

Dear Dr. Sha'aban,

We look forward to receiving your revised manuscript.

Kind regards,

Antar S. H. Abdul-Qawy, Ph.D.

Academic Editor

PLOS ONE

Journal Requirements:

Additional Editor Comments:

**Dear authors, **

**Even though some reviewers have confirmed that their concerns have been addressed, one reviewer has continued to raise that some of his comments have not been adequately addressed, while another has noted issues regarding the clarity of your contributions, algorithm, and results presentation.**

**After carefully assessing the revised manuscript and your responses to all reviewers’ comments, I believe these concerns are valid, and I am not fully satisfied with the level and clarity of your response. Therefore, I am issuing another revision decision and require that all the concerns be adequately, clearly, and properly addressed. If these concerns remain unaddressed or only partially resolved, I may reject the manuscript.**

Reviewer's Responses to Questions

**Comments to the Author**

Reviewer #1: All comments have been addressed

Reviewer #2: All comments have been addressed

Reviewer #3: All comments have been addressed

Reviewer #4: All comments have been addressed

Reviewer #5: All comments have been addressed

Reviewer #6: (No Response)

Reviewer #7: (No Response)

2. Is the manuscript technically sound, and do the data support the conclusions?

Reviewer #1: Yes

Reviewer #2: Yes

Reviewer #3: Yes

Reviewer #4: Yes

Reviewer #5: Yes

Reviewer #6: Partly

Reviewer #7: Partly

3. Has the statistical analysis been performed appropriately and rigorously?

Reviewer #1: N/A

Reviewer #2: Yes

Reviewer #3: Yes

Reviewer #4: Yes

Reviewer #5: N/A

Reviewer #6: No

Reviewer #7: N/A

4. Have the authors made all data underlying the findings in their manuscript fully available?

Reviewer #1: Yes

Reviewer #2: Yes

Reviewer #3: Yes

Reviewer #4: Yes

Reviewer #5: Yes

Reviewer #6: Yes

Reviewer #7: No

5. Is the manuscript presented in an intelligible fashion and written in standard English?

Reviewer #1: Yes

Reviewer #2: Yes

Reviewer #3: (No Response)

Reviewer #4: Yes

Reviewer #5: Yes

Reviewer #6: Yes

Reviewer #7: Yes

Reviewer #1: The authors have correctly addressed all the comments. I have no additional remarks.

The authors have correctly addressed all the comments. I have no additional remarks.

Reviewer #2: now the manuscript is in acceptable format hence,after some minor revision the manuscript is acceptable

Reviewer #3: The authors have addressed the comments, and the quality of the paper has improved. I therefore accept the publication of this paper in its current form.

Reviewer #4: The current version is more refined as compared to the last time. The implementations scenarios have been addressed better in detail. However there are typos in the current version that needs to be revised by the authors. Just a sample I am sharing

Consider a farm with N sensor nodes and M UAVs operating at 200*00 2..... what is this dimension? there seems to be some error in it. These kind of technical typos need attention.

Reviewer #5: Thank you for effectively addressing all the previous comments. The paper is now in a better form and hence acceptable for publication.

Reviewer #6: The authors introduce Sky and Wire, a synchronization algorithm that integrates Unmanned Aerial Vehicles with Broadband over Power Line (UAV-BPL) for herding operations. Two scenarios are analyzed: Scenario 1, the coverage-driven scenario and Scenario 2, the priority-driven exchange scenario. The algorithms corresponding to these scenarios are validated through MATLAB simulations.

Comments on the document:

Although the proposed UAV-BPL framework is very interesting and offers significant advantages, the document does not clearly explain how the proposed solution works. This lack of clarity is critical, since the main contributions of the paper are the algorithms themselves. A more detailed and consistent explanation is needed.

For example, in Section 3.2, Phase 3, it is stated that the UAV transmits data collected from sensors to a nearby relay, which then forwards the data to cloud storage via BPL. However, just before, in Phase 2, the text states that data packets Pi are collected by the UAV either directly from sensors or FROM its designated RELAY, depending on the priority. It is therefore unclear why the UAV both collects data from relay points and transmits data to them.

Figures 1–3 show no modifications; no improvements have been made to them (see previous comments).

In Equations (3) and (4), the symbols “x” and “ . ” appear, but their meaning is not explained. If both are simply multiplication signs, why the difference?.

The definition of d_{ij} following Equation (3) has not been corrected.

The parameters used in all simulations should be clearly presented. What are the "random seeds" used for the presented results?

The presentation of results requires improvement (e.g., fonts, axis labels). In particular, the differences in UAV trajectories should be made more noticeable.

Reviewer #7: Thank you for the revised manuscript and your responses. I appreciate the effort. However, many replies are brief and general, and they do not point to the exact places in the manuscript where changes were made. The paper still needs substantial refinement. Please treat the reviewers’ suggestions as an opportunity to improve the work.

How to respond going forward: when you write “this has been addressed/clarified,” please name the exact Section, Page, and Line numbers so the reviewer can verify the change quickly.

The following points refer to my earlier comments that have not yet been adequately addressed:

1. Comment #11 (notation xᵢⱼ(t) vs dᵢⱼ(t)): In revised manuscript, Section 3.3.1 (p. 9, lines 12–13) you write that dᵢⱼ(t) is the binary decision variable for UAV movement (Euclidean distance). This is inconsistent. Is dᵢⱼ(t) a distance (a real value) or a binary decision variable? If it is binary, how does it differ from xᵢⱼ(t)? Please clarify the definitions, use them consistently in text and equations, and ensure the paper reflects this correction.

2. Comment #14: I am not satisfied with the current response. Please revisit the original point and address it fully in the manuscript, indicating where changes were made (Section/Page/Line).

3. Comments #17 and #22 (references): I could not locate the sources you cited. Please provide precise citations (Section/Page/Line) and list the full references you used for the “quadcopter” claim.

4. Comment #10 (“critical data sources”): I could not find a clear answer. Please respond directly to this point in your rebuttal and update the manuscript accordingly, with exact locations of the changes.

5. Comment #20 (data path inconsistency): Your response says UAVs collect data from BPL relay points (aggregated sources), but the revised manuscript (Step 1 in your synchronization algorithm, p. 8) says UAVs collect directly from a sensor when priority pᵢ(t) > θ. Please reconcile this: under what conditions do UAVs collect from relays vs directly from sensors? Align the algorithm and discussion so they tell one consistent story.

6. Comment #22 (runs vs iterations): You wrote “a multi-iteration simulation spanning 10 runs.” Please clarify whether this means 10 runs (independent trials with results averaged) or 10 iterations within one run (time steps within a single simulation). Remember that they are different. Define "run" and "iteration," what varies and what is fixed in your simulation, and reflect this clearly in the methods and results discussion.

**Do you want your identity to be public for this peer review?** For information about this choice, including consent withdrawal, please see our Privacy Policy

Reviewer #1: No

Reviewer #2: No

Reviewer #3: No

Reviewer #4: No

Reviewer #5: **Yes: ** Dr. Vincent Omollo Nyangaresi

Reviewer #6: No

Reviewer #7: No

---

## [Author Response · Author response to Decision Letter 2]

27 Sep 2025

We would like to thank the Editorial team and anonymous reviewers for their feedback and comments, which helped improve this work. We have reviewed the reviewer’s comments to clarify them and make necessary corrections.

---

## [Editor Report · Decision Letter 2]

6 Oct 2025

SKY AND WIRE: A UAV-BPL SYNCHRONIZATION ALGORITHM FOR INFORMATION EXCHANGE IN HERDING OPERATIONS

PONE-D-25-35141R2

Dear Dr. Sha'aban,

We’re pleased to inform you that your manuscript has been judged scientifically suitable for publication and will be formally accepted for publication once it meets all outstanding technical requirements.

Kind regards,

Antar S. H. Abdul-Qawy, Ph.D.

Academic Editor

PLOS ONE

Additional Editor Comments (optional):

Please note that I have acted as a reviewer for this manuscript, and you will find my comments under Reviewer 7
---

## [Editor Report · Acceptance letter]

PONE-D-25-35141R2

PLOS ONE

Dear Dr. Sha'aban,

I'm pleased to inform you that your manuscript has been deemed suitable for publication in PLOS ONE. Congratulations! Your manuscript is now being handed over to our production team.

Kind regards,

on behalf of

Dr. Antar S. H. Abdul-Qawy

Academic Editor

PLOS ONE